A Sarcoptes scabiei specific isothermal amplification assay for detection of this important ectoparasite of wombats and other animals

Fraser Tamieka A. 1 2
Carver Scott 2
Martin Alynn M. 2
Mounsey Kate 1 3
Polkinghorne Adam 1
Jelocnik Martina Martina.Jelocnik@research.usc.edu.au 1
1 USC Animal Research Centre, Faculty of Science, Health, Education and Engineering, University of the Sunshine Coast , Sippy Downs , Australia
2 Department of Biological Sciences, University of Tasmania , Sandy Bay , Australia
3 School of Health and Sport Sciences, University of the Sunshine Coast , Sippy Downs , Australia
Gutiérrez Carlos
Electronic publication date: 2018 Jul 27
Publication date: 2018
Volume: 6
Electronic Location ID: e5291
Received 2018 Feb 28; Accepted 2018 Jul 2
Copyright: ©2018 Fraser et al.
Copyright year: 2018
Copyright holder: Fraser et al.
License: This is an open access article distributed under the terms of the Creative Commons Attribution License, which permits unrestricted use, distribution, reproduction and adaptation in any medium and for any purpose provided that it is properly attributed. For attribution, the original author(s), title, publication source (PeerJ) and either DOI or URL of the article must be cited.
License URL: https://creativecommons.org/licenses/by/4.0/

Keywords: LAMP, Diagnostics, Sarcoptic mange, Skin scraping, PCR, One health, Australian wildlife, Sarcoptes scabiei, Wombats

Funding: Holsworth Wildlife Research Endowment award This work was supported by a Holsworth Wildlife Research Endowment awarded to Tamieka Fraser. The funders had no role in study design, data collection and analysis, decision to publish, or preparation of the manuscript.

==============================
Background

The globally distributed epidermal ectoparasite, Sarcoptes scabiei, is a serious health and welfare burden to at-risk human and animal populations. Rapid and sensitive detection of S. scabiei infestation is critical for intervention strategies. While direct microscopy of skin scrapings is a widely utilised diagnostic method, it has low sensitivity. PCR, alternatively, has been shown to readily detect mite DNA even in microscopy-negative skin scrapings. However, a limitation to the latter method is the requirements for specialised equipment and reagents. Such resources may not be readily available in regional or remote clinical settings and are an important consideration in diagnosis of this parasitic disease.

Methodology

A Loop Mediated Isothermal Amplification (LAMP) assay targeting the ITS-2 gene for S. scabiei was developed and evaluated on clinical samples from various hosts, previously screened with conventional S. scabies-specific PCR. Species specificity of the newly developed LAMP assay was tested against a range of DNA samples from other arthropods. The LAMP assays were performed on a real-time fluorometer as well as thermal cycler to evaluate an end-point of detection. Using skin scrapings, a rapid sample processing method was assessed to eliminate extensive processing times involved with DNA extractions prior to diagnostic assays, including LAMP.

Results

The S. scabiei LAMP assay was demonstrated to be species-specific and able to detect DNA extracted from a single mite within a skin scraping in under 30 minutes. Application of this assay to DNA extracts from skin scrapings taken from a range of hosts revealed 92.3% congruence (with 92.50% specificity and 100% sensitivity) to the conventional PCR detection of S. scabiei. Preliminary results have indicated that diagnostic outcome from rapidly processed dry skin scrapings using our newly developed LAMP is possible in approximately 40 minutes.

Discussion

We have developed a novel, rapid and robust molecular assay for detecting S. scabiei infesting humans and animals. Based on these findings, we anticipate that this assay will serve an important role as an ancillary diagnostic tool at the point-of-care, complementing existing diagnostic protocols for S. scabiei.

Introduction

Sarcoptes scabiei is an ectoparasite that resides in the epidermal layer of its hosts causing a range of clinical signs of disease including pruritis, dermal inflammation, hyperkeratosis and alopecia, which may lead to bacterial sepsis (McCarthy et al., 2004). S. scabiei is listed among the top 50 most prevalent diseases in humans with over 100 million people globally predicted to be infested (Hay et al., 2014; Romani et al., 2015). Beyond its role in human disease, a wide range of domestic animals, wild canids, and other wildlife suffer extensively from sarcoptic mange, and transmission to at-risk animal populations can result in population declines and localised extinctions (Forchhammer & Asferg, 2000; Gakuya et al., 2011; Graczyk et al., 2001; Martin et al., 2018; Perrucci et al., 2016). With the endemic infestation of humans in tropical and subtropical areas, the large variety of animal species infested and the knowledge that S. scabiei is the same mite infesting all, pathogen dispersal and spill-over has been suggested to be the causative consequence of global infestations (Fraser et al., 2016; Walton & Currie, 2007).

As with many infectious diseases, the successful treatment of affected individuals and the application of appropriate disease management strategies relies on rapid and accurate detection of the infectious agent. Diagnosis of scabies (also classified as mange in animals) is typically made by assessment of clinical features alone (Hardy, Engelman & Steer, 2017; Walton & Currie, 2007). When atypical appearances are presented, however, the diagnosis can be challenging as other skin conditions can mimic clinical signs of scabies (Hardy, Engelman & Steer, 2017). A skin scraping of the affected area provides a more definitive diagnosis as mites, mite eggs and faecal pellets can be identified via microscopy due to their distinct morphology (Leung & Miller, 2011). Although the diagnosis is more specific using microscopy, detection of early mange has been shown to have limited sensitivity, primarily due to the difficulties in sampling and visualising mites when the mite burden is low (Fraser et al., 2018; Skerratt, 2005; Walton & Currie, 2007). Recent studies have shown that diagnosis of S. scabiei by clinical features and microscopy are unreliable methods for early stage infestations (Fraser et al., 2018; Wong et al., 2015).

Besides microscopy, alternative diagnostic methods for S. scabiei have been evaluated with varying sensitivity and specificity. Several studies have attempted to use serological techniques (i.e., ELISAs) as a more targeted diagnostic method (Arlian, Feldmeier & Morgan, 2015; Löwenstein, Kahlbacher & Peschke, 2004; Rambozzi et al., 2004; Rodríguez-Cadenas et al., 2010; Zhao et al., 2014). However, as reviewed by Arlian & Morgan (2017), significant limitations for this method exist including the time taken for the host to develop S. scabiei-specific antibodies and cross-reactivity between S. scabiei antigens and those from other mites. Molecular techniques using nucleic acid amplification tests (NAATs) as a diagnostic tool for S. scabiei are relatively new but show promising results (Fraser et al., 2018; Wong et al., 2015). Two studies, analysing samples collected from humans (Wong et al., 2015) and animals (Fraser et al., 2018), have demonstrated that PCRs have a higher sensitivity and specificity than microscopy, revealing high rates of false negative samples previously screened by microscopy. However, NAATs are not well adapted for clinical settings, particularly for diseases like scabies which are common in remote or resource-limited communities and/or in field settings with limited access to diagnostic laboratories and necessary equipment (Walton & Currie, 2007). Recent advances in this field have utilised hand held devices resulting in promising outcomes for detecting infectious diseases quickly. This includes the Biomeme Inc. portable PCR machine with thermocycler and fluorometer which can dock into an iPhone resulting in rapid results and the AmplifyRP® portable florescence reader (Marx, 2015; Zhang et al., 2014).

Loop mediated isothermal amplification (LAMP) is one of the expanding range of NAAT techniques that is showing capacity at the Point of Care (POC). LAMP assays are low cost, rapid and can be used with simple ‘bench-top’ equipment. Visual result interpretation with the use of different DNA binding dyes in these assays further support LAMP use at the POC. There have been multiple successful LAMP assays developed for other human and veterinary parasites including Plasmodium spp. (Lucchi et al., 2016), Toxoplasma gondii (Kong et al., 2012) and Leishmania spp. (Adams et al., 2010), and bacteria including Chlamydia spp. (Jelocnik et al., 2017), Mycoplasma pneumoniae (Saito et al., 2005) and Streptococcus agalactiae (McKenna et al., 2017). To overcome on some limitations associated with LAMP assays, including misleading of results using turbidity techniques and cross contamination as a result of opening tubes, the use of a fluorometer and a signature melt for amplicon characteristics can account for these limitations.

This study aimed to develop a LAMP assay for the detection of S. scabiei in animals and assess its reliability against PCR and demonstrate its potential as a POC test. We have utilised a unique sample set of skin scrapings taken from a range of hosts and tested extracted DNA from those with the newly developed S. scabiei specific LAMP assay. The LAMP assay was evaluated against microscopy and a recently described conventional S. scabiei-specific PCR assay. In an attempt to reduce sample processing time, we also optimised a rapid DNA extraction method on a small subset of skin scrapings, further highlighting the potential for this assay to be deployed at the POC.

Methods and Materials

LAMP assay design

The S. scabiei internal transcribed spacer 2 (ITS-2) gene is a highly conserved gene and was used as the LAMP target in this study. A ClustalW alignment of 87 ITS-2 sequences (represented as haplotypes) from S. scabiei mites from humans and a variety of animals across Australia, Europe, North America and Asia available in GenBank was obtained to identify polymorphisms in this gene (Fig. S1, Table S1). In addition, we have included ITS-2 sequences from other mite species, including the house dust mite (Dermatophagoides farinae), the chorioptic mange mite (Chorioptes sp), the notoedric mange mite (Notoedres cati), the psoroptic mange mite (Psoroptes sp) and ticks (Ixodes sp). This 450 bp fragment was also subjected to a discontiguous megablast search in Basic Local Alignment Search Tool (BLAST) (http://blast.ncbi.nlm.nih.gov/Blast.cgi#) (Johnson et al., 2008) to evaluate S. scabiei sequence specificity. Primer explorer V5 (Eiken Chemical Co., Tokyo, Japan) was used for primer design and yielded five sets of primers consisting of two outer (F3 and B3) and two inner (FIP and BIP) primers. These sets were analysed and three were excluded due to sequence overlap and primer parameters. The remaining two sets were selected for further testing with both sets analysed in silico with BLAST (Johnson et al., 2008) and the OligoAnalyzer 3.1 online tool (https://sg.idtdna.com/calc/analyzer) (Owczarzy et al., 2008) to assess primers for DNA base mismatches, hairpins and annealing temperature. Loop primers were additionally designed manually to increase sensitivity and accelerate the reaction time (Fig. 1, Table 1).

Figure 1 S. scabiei. LAMP primer sequences.

Two outer (F3 and B3), two inner (FIP and BIP) and two looping (LF and LB) primers for S. scabiei specific LAMP assay outlined on the ITS-2 sequence (Genbank accession number AB778896).

Table 1 LAMP primers used in this study.

Name	Sequence 5′-3′	Position	Length	
F3	TGTTAGTAGTAGCTCTATGAGAA	148–170	23	
B3	TCGCTTGATCTGAGGTCG	364–347	18	
FIP (FiC + F2)	ACCCTAGGAGAATGTCGCACAATGTTTCAAGTCTCGAGTGG		41	
BIP (BiC + B2)	CAGTGATGTGTGCCTGTTGAGAGAAATGACATTTCATTGCTTGT		44	
Loop F	CATCGATGTGCTTTCAA	210–194	17	
Loop B	CATGAATATCAAAGAGTG	301–318	18	
F2	AATGTTTCAAGTCTCGAGTGG	171–191	21	
FiC	ACCCTAGGAGAATGTCGCAC	230–211	20	
B2	CAGTGATGTGTGCCTGTTGAGA	345–324	22	
BiC	GAAATGACATTTCATTGCTTGT	264–285	22	

S. scabiei LAMP assay validation

Each LAMP reaction consisted of a 15 µL Isothermal Master Mix ISO001 (Optigene, Horsham, UK), 5 µL primer mix (at 0.2 µM F3 and B3, 0.8 µM FIP and BIP, and 0.4 µM LF and LB) and 5 µL of DNA template. Initial testing and validation of the two primer sets was performed on S. scabiei DNA from a single mite and a pool of three mites at 65 °C for 30 min using a heating block, with results visualised on an ethidium bromide agarose gel under UV light. During this development step, the second primer set was excluded due to high primer dimerization (not shown), and LAMP primers described in Fig. 1 and Table 1 were used henceforth. Confirmation of the LAMP target sequence was completed by sequencing the amplification product generated with outer F3 and B3 primers, with the resulting sequence deposited in Genbank under accession number MH379093.

After initial optimisation, samples were tested using the Genie III real-time fluorometer (Optigene, Horsham, UK), and reactions were run at 65 °C for 30 min, followed by annealing at 98 °C to 80 °C at a rate of 0.05 °C/s to generate the signature melt profile (curve) of the amplified product. A negative control consisting of water as template was included in each run.

LAMP gene target specificity was evaluated using other arthropod DNA (Pediculus humanus, Leptotrombidium pallidum, Periplaneta australasiae, Bovicola ovis, Bovicola bovis, Solonopotes capillatus, Ixodes holocyclus, Ixodes tasmani), and skin scrapings negative for S. scabiei (as previously determined by PCR and microscopy Fraser et al., 2018).

Clinical samples used in this study

The evaluation of the S. scabiei LAMP assay was performed on (i) DNA extracts from 40 skin scrapings collected from 23 wombats (Vombatus ursinus) as previously described (Fraser et al., 2018) and, (ii) 24 DNA extracts from individual skin scrapings collected from five domestic dogs (Canis lupus familiaris), eight wombats, two koalas (Phascolarctos cinereus), two wallabies (Macropodidae sp.), and seven known healthy wombats, stored in 80% ethanol at −80 °C (Table S2). The DNA extraction procedure was performed as previously described using QiaAMP DNA Mini kit (Qiagen, Valencia, CA, USA) (Fraser et al., 2018). The collection and use of these samples was approved by the Animal Research Committee at the University of the Sunshine Coast (approval AN/S/16/43, and AN/E/17/17), the Animal Research Committee at the University of Tasmania (approval A0014670) and state permits from Office of Environment & Heritage NSW National Parks & wildlife Service (SL101719), Department of Primary Industries, Park, Water and Environment for Tasmania (approval FA15121) and The Department of Environment, Land, Water and Planning for Victoria (10007943). All methods were carried out in accordance with the 2013 Australian National Health and Medical Research Council ‘Australian code for the care and use of animals for scientific purposes’. Aforementioned samples were also screened by conventional PCR targeting a 374 bp fragment of the S. scabiei cox1 gene, having a respective sensitivity and specificity of 100% and 84.62% in concordance to microscopy, as previously described (Fraser et al., 2018). PCR positivity for cox1 was determined by visualisation of the 374bp fragment following agarose gel electrophoresis under UV light.

In order to confirm negative samples and to test for isothermal amplification inhibition, a subset of six negative samples were spiked with 10 µL mite only DNA and tested again by LAMP.

Evaluation of rapid skin scraping DNA extraction

In order to assess the use of LAMP at the POC, eleven wombat skin scrapings, with mite counts previously assessed by microscopy, were submerged with 0.3M Potassium Hydroxide (KOH), pH 13, and heated at 95 °C for 10 min in order to lyse the tissue and release DNA from the cells. After vortexing, 5 µL of the tissue suspension was used as a template in each reaction, also consisting of 15 µL of Lyse’n’Lamp master mix (Optigene, Horsham, UK) and 5 µL primer mix as described above. The LAMP reaction was performed in the Genie III fluorometer using the same cycling conditions as described above. Negative controls of water only and an aliquot of 0.3M KOH only were included in the assays. The same samples were also tested with the cox 1 PCR after performing DNA extractions, as described above, on the KOH skin suspensions.

Statistical analysis

The performance of the LAMP assay compared to the reference PCR assay conducted on the same samples was estimated by calculating Kappa values, overall agreement, sensitivity and specificity. Direct comparisons were conducted using EpiTools online (http://epitools.ausvet.com.au) (Sergeant, 2017). Kappa values are interpreted as follows: values ≤0 as indicating no agreement, 0.01–0.20 as none to slight, 0.21–0.40 as fair, 0.41–0.60 as moderate, 0.61–0.80 as substantial, and 0.81–1.00 as almost perfect agreement.

Results

S. scabiei LAMP assay development and validation

The LAMP primers were predicted to amplify a 217 bp fragment of the ITS-2 gene. The alignment of the available S. scabiei ITS-2 gene sequences (n = 87) revealed 96.5%–100% sequence identity (Fig. S1). ”-In silico analysis of ITS-2 sequences obtained from Dermatophagoides farinae (KT724354), Chorioptes sp. (AF123084), Ixodes pavlovskyi (KP242014), Ixodes persulcatus (KR136379), Notoedres cati (AF251801), Psoroptes natalensis (AB968091), Psoroptes cuniculi (KP676689) and Psoroptes ovis (EF429259) indicated that the S. scabiei LAMP primers are likely to be specific, as we identified 101 to 239 nucleotide polymorphisms between our primers and other arthropod sequences (Fig. S1).

Figure 2 Amplification and melt outputs for S. scabiei using specific isothermal amplification.

Outputs from the LAMP experimental run; (A) showing amplification and (B) melt outputs using both positive and negative samples in the assay. A water as a template and single mite DNA (Ss2) were included as negative and positive control in the run. Samples with melt at 85 °C are deemed positive.

The S. scabiei LAMP assay was initially assessed in house on a thermal block, with the assay run for 30 min at 65 °C. A single mite as well as pooled mite DNA extracts were detectable by LAMP, as visualised by the amplicons on the gel. We also tested 10-fold dilutions of a mite and mite-positive skin scraping DNA samples by LAMP on the thermal cycler in two independent runs using the same run conditions (Fig. S2). We successfully amplified 10−2 single mite DNA dilutions (0.02 ng/µL of DNA) and 10−4 skin scraping DNA dilutions (0.008 ng/µL of DNA) (Fig. S2).

When the LAMP assay was run in the Genie III fluorometer (Optigene, Horsham, UK) for 30 min at 65 C, DNA extracted from a single mite (Ss2) and a pool of three mites (Ss3) resulted in amplification times of 20.00 and 11.30 min, respectively, with melt temperatures ranging between 85.26 °C to 85.39 °C (Fig. 2, Table S2). Additional validation of species specificity of the S. scabiei LAMP assay was performed on a panel of DNA extracts from other arthropods and S. scabiei negative skin scrapings on both thermal block and real time fluorometer. None of the other arthropod DNA and previously validated S.  scabiei-negative skin scraping samples produced LAMP amplification. In comparison to using thermal block for incubation, we found that S. scabies LAMP assays run in the Genie III fluorometer are less laborious and the chance of contamination is decreased as the amplification product is confirmed with its signature melt and is within a closed system.

Validation of the S. scabiei LAMP on clinical specimens

A total of 64 clinical samples were tested by the newly developed scabies LAMP, as well as the S. scabiei cox1 PCR assay. A high congruence of 95.3% (61/64) was found between the S. scabiei LAMP assay and conventional S.  scabiei PCR assay, previously shown to be more sensitive than microscopy (Fraser et al., 2018), with 24 positive and 37 negative samples in agreement (Table 2). Only three samples were positive by S. scabiei LAMP but negative by conventional PCR. Overall, LAMP had a sensitivity and specificity of 100% (Clopper–Pearson 95% CI [0.86–1.00]) and 92.50% (Clopper–Pearson 95% CI [0.80–0.98]) respectively when compared to the PCR assay (Table 2). Kappa of 0.90 (95% CI [0.84–1.01]) indicated a near perfect agreement between LAMP and the conventional S. scabiei cox1 PCR in this study. In order to confirm negative results, six negative samples were “spiked” with mite DNA. All six “spiked” samples produced amplification times of 11.15–16.15 min with melts of 85.28–85.67 °C, suggesting that potential inhibitors in the system did not prevent amplification from occurring (Table S3).

Table 2 Comparison of the S. scabiei LAMP and PCR assays for clinical skin scraping DNA.

PCR	LAMP		Kappar (95% CI)	Sensitivity (95% CI)	Specificity (95% CI)	
	Positive	Negative	Total				
Positive	24	0	24	0.90 (80–101)	100% (86–100)	92.50% (80–98)	
Negative	3	37	40	
Total	27	37	64	

Reproducibility

To assess the reproducibility of the LAMP assay, a repeat set of 14 samples were selected and run in triplicate by two operators using blind testing principles (Table S4). Assays showed reproducibility of both positive and negative results with small variation between amplification times and melt. Amplification times for each sample varied between 0–3 min and 0.05–0.5 °C in melt for each sample.

Rapid specimen processing

We also evaluated the use of an Optigene Lyse’n’Lamp isothermal master mix for rapid DNA extraction prior to S. scabiei LAMP. Application of this step to a panel of S. scabiei positive (n = 7) and negative (n = 4) wombat skin scrapings (determined by microscopy), revealed 100% congruence to the microscopy result (Table 3) with amplicons generated in positive samples with a time range of 13.2–26.0 min and melts of 84.28–85.20 °C. As previously noted by Fraser et al. (2018), microscopy is not always the most reliable method. Therefore, the assessment between conventional PCR and LAMP was also assessed. There was an 80% congruence between the two tests (8/10) with a single sample positive by PCR but negative by LAMP (C3), a single sample negative by PCR but positive by LAMP (A1). Only one sample (A5) did not have sufficient volume to complete a DNA extraction and subsequent conventional PCR.

Table 3 LAMP, microscopy and PCR results of the rapidly processed 11 wombat skin scrapings.

Sample	Microscopy	PCR	LAMP	Time to amplify (minutes)	Melt (°C)	
A1	Positive	Negative	Positive	17:15	84.96	
A3	Positive	Positive	Positive	22:15	84.91	
A5	Positive	NA	Positive	16:00	84.65	
A7	Positive	Positive	Positive	13:15	85.20	
B1	Negative	Negative	Negative			
B3	Positive	Positive	Positive	18:15	84.61	
B5	Positive	Positive	Positive	26:00	84.28	
B7	Negative	Negative	Negative			
C1	Positive	Positive	Positive	22:30	85.07	
C3	Negative	Positive	Negative			
C5	Negative	Negative	Negative			
Notes.

NA not applicable

Discussion

The accurate and rapid detection of S. scabiei in clinical specimens is critical to the appropriate and timely treatment of affected individuals and the implementation of control strategies to reduce the transmission of this parasitic mite. Microscopic examination of skin scrapings continues to be the gold standard for the detection of S. scabiei in combination with a detailed clinical assessment. However, comparisons with molecular methods have revealed shortcomings in the sensitivity in the detection of this ectoparasite, particularly in individuals with only low S. scabiei mite counts (Fraser et al., 2018; Hardy, Engelman & Steer, 2017). Building on a growing evidence for the diagnostic utility of NAATs, this study successfully developed a rapid and specific LAMP assay as an ancillary method for detection of S. scabiei at the POC.

We demonstrated that the newly designed S. scabiei LAMP assay is not just specific and robust, but also capable of providing a rapid diagnostic result (within 30 min). However, much like microscopy and PCR, the result of LAMP is only a reflection within the sample itself rather than the overall health of the individual, the amount of DNA containing a mite in a typical skin scraping does lie to chance. Beyond the increase in speed of this new S. scabiei assay compared to a conventional PCR assay, we show that DNA from a single mite can be easily detected in a clinical sample. This level of sensitivity is particularly important given that (a) the mite load is highly variable in individuals compared to their disease presentation, particularly in individuals with early signs of scabies/mange (Arlian & Morgan, 2017); and (b) the quality and size of the skin scraping is likely to significantly affect the number of mites that will be sampled from the dermis at the affected anatomical site (Fraser et al., 2018). It has been previously described that even multiple skin scrapings from the same region will harbour varying results for sarcoptic mange. This is not limited to scabies alone, as low parasitic burdens complicate sampling and diagnosis of other parasitic infestations, such as cutaneous leishmaniasis (CL), particularly in post-treatment (Faber et al., 2003). LAMP assays on skin biopsies of CL patients were found to be successful prior to treatment, but were unsuccessful in follow-up collections as a result of a low parasite burden (Adams et al., 2010). Hence, the sensitivity of detection reported in this newly designed S. scabiei LAMP assay is promising. Indeed, the detection of S. scabiei DNA in several samples that were negative by our comparative S. scabiei conventional PCR assay would suggest that LAMP could be even more sensitive than conventional PCR for S. scabiei detection. Potential explanations for this enhanced sensitivity include that (i) the quantity of amplicons generated by LAMP assays are considerably higher in comparison to those produced by PCR, (ii) due to the use of six specific primers rather than two used in PCR the amplification itself is more targeted, and (iii) the detection of amplicons is by fluorescence rather than visualisation on an agarose gel (Notomi et al., 2000).

A critical aspect limiting the application of this NAAT assay at the POC is the general requirement for a DNA extraction prior to PCR detection. Although our results are preliminary, our data showed that rapid commercial DNA processing kits such as those used in this study appear to be effective in lysing mites embedded in tissues to release DNA for detection. When this method was combined with the LAMP assay itself, it meant that the complete reaction time from sampling to result would be generally around 40 min. Such innovation, alongside alternative sample processing and amplicon detection methods, has the potential to make this S. scabiei LAMP assay a POC reality with further development. We and others have also investigated the use of swab sampling of the affected dermis for S. scabiei detection (Fraser et al., 2018; Wong et al., 2015). If further validation would reveal that this non-invasive sampling is suitable for S. scabiei detection, this approach combined with the LAMP assay would fulfil the requirements for an assay that can be deployed in a range of clinical settings.

In this study we focussed on the ITS-2 gene as a LAMP target, however three S. scabiei mitochondrial genes cox1, 12S rRNA and 16S rRNA could be potentially viable targets for LAMP assays based on the availability of sequence data. While not investigated here, we considered but did not pursue these less viable options, as (i) the sequence variation is higher at these mitochondrial loci across the different S. scabiei evolutionary lineages and (ii) low GC% content of mitochondrial genes could be problematic for the LAMP assay primer design (Fraser et al., 2016; Fraser et al., 2017). Unfortunately, in this study we did not have human S. scabiei samples to evaluate with our new assay. Nevertheless, our research (Fraser et al., 2017) and that of others (Alasaad et al., 2009; Amer et al., 2014; Andriantsoanirina et al., 2015; Gu & Yang, 2008; Zahler et al., 1999) shows ITS-2 to be highly conserved across host species and, thus, there is no evidence to suggest the LAMP assay developed here would not work equally well on S. scabiei infesting humans.

Besides detection efficiency and species-specific target, other factors (including cost, time and technical skill) should be also considered during the development of a new diagnostic assay. Regarding scabies, microscopy, although cost efficient, requires technical skill to distinguish mites and eggs within a skin scraping, with a single scraping analysed at a time. Conversely, PCR assays can analyse multiple samples at a time and are highly sensitive at detecting mite DNA, eliminating the requirement for specialised technical skills. However, the time to obtain a result can take up to 24 h and the additional equipment and reagents (i.e., accessible laboratory, PCR and electrophoresis equipment and DNA extraction kits) are essential for this diagnostic method. The LAMP assay, in comparison, can be both cost efficient and rapid, requiring general technical expertise. The fluorometer (such as Genie III used in this study), as one-off cost all-inclusive instrument is portable and can be run off batteries. As the master mix can be pre-aliquoted and the template rapidly processed, the requirement for additional kits and laboratory space is limited. However, further development for a microfluidic device or colorimetry and the efficiency of the rapid DNA extraction is required for this LAMP assay to be classified as a POC test.

Conclusion

In conclusion, this study describes a development of a new assay for the animal and human S. scabiei detection at the POC as well as laboratory. With further development, this assay has the potential to complement existing diagnostic methods in the clinical setting and may offer a low cost, portable option for S. scabiei DNA detection in remote or resource-deficient regions.

Supplemental Information

Table S1 Accession numbers of the 87 ITS-2 sequences obtained from GenBank for alignment

Click here for additional data file.

Table S2 Results for microscopy mite count, PCR and LAMP for all samples used in this study

Each sample has a corresponding host, microscopy mite count, skin scraping PCR and LAMP result. In some instances, multiple skin scrapings were taken from an individual, reflected by the sample name_body location. LF: left flank, RF: right flank, N: neck, H: head, RA: right forearm, RL: right hind leg, B: back, LL: left hind leg. If multiple skin scrapings were taken from the same body location this is described by the number next to name_body location.

Click here for additional data file.

Table S3 Spiking results of negative samples to eliminate LAMP inhibition

Click here for additional data file.

Table S4 Reproducibility of S. scabiei LAMP using skin scraping DNA extracts and mite only DNA

Click here for additional data file.

Figure S1 Alignment of the available S. scabiei and other arthropod ITS-2 gene sequences from GenBank

Click here for additional data file.

Figure S2 End-point detection of S. scabiei LAMP performed on the thermal cycle using 10-fold serial dilutions of single mite and wombat skin DNA

UV visualisation of a single mite (Ss2: 2.8 ng/µL) and a S. scabiei positive wombat skin scraping (DW02_1: 86 ng/µL) LAMP amplicons on an ethidium bromide stained agarose gel. DNA serial dilution LAMP assays were performed on a thermal block for 30 minutes at 65 °C. Two negative controls containing water were also included. DNA Molecular Weight Marker VIII (Sigma-Aldrich) was used.

Click here for additional data file.

We would like to thank Dr. Renfu Shao and Ms. Delaney Burnard for supplying non-S. scabiei DNA to assist with assessing the specificity of our assays. We would also like to thank Dr. Cam Raw, Ms. Roz Holme, Ms. Michelle Thomas and Ms. Merridy Montarello for supplying additional clinical skin scrapings for molecular typing, which we used opportunistically for the current LAMP assay.

Additional Information and Declarations

Competing Interests

Author Contributions

Animal Ethics

DNA Deposition

Data Availability

The authors declare there are no competing interests.

Tamieka A. Fraser conceived and designed the experiments, performed the experiments, analyzed the data, contributed reagents/materials/analysis tools, prepared figures and/or tables, authored or reviewed drafts of the paper, approved the final draft.

Scott Carver analyzed the data, contributed reagents/materials/analysis tools, authored or reviewed drafts of the paper, approved the final draft.

Alynn M. Martin and Kate Mounsey performed the experiments, contributed reagents/materials/analysis tools, approved the final draft.

Adam Polkinghorne analyzed the data, contributed reagents/materials/analysis tools, authored or reviewed drafts of the paper, approved the final draft.

Martina Jelocnik conceived and designed the experiments, analyzed the data, contributed reagents/materials/analysis tools, prepared figures and/or tables, authored or reviewed drafts of the paper, approved the final draft.

The following information was supplied relating to ethical approvals (i.e., approving body and any reference numbers):

The collection and use of these samples was approved by the Animal Research Committee at the University of the Sunshine Coast (approval AN/S/16/43, and AN/E/17/17), the Animal Research Committee at the University of Tasmania (approval A0014670) and state permits from Office of Environment & Heritage NSW National Parks & Wildlife Service (SL101719), Department of Primary Industries, Park, Water and Environment for Tasmania (approval FA15121) and the Department of Environment, Land, Water and Planning for Victoria (10007943). All methods were carried out in accordance with the 2013 Australian National Health and Medical Research Council ‘Australian code for the care and use of animals for scientific purposes’.

The following information was supplied regarding the deposition of DNA sequences:

The 217 bp ITS gene fragment, described in this study is deposited in Genbank under accession number MH379093.

The following information was supplied regarding data availability:

The raw data are provided in Table S2.

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
