# Peer review of "A Sarcoptes scabiei specific isothermal amplification assay for detection of this important ectoparasite of wombats and other animals"

_PeerJ, doi:10.7717/peerj.5291_

## Round 0.1 · original submission · Major Revisions

In my opinion the manuscript is relevant and well written. From my side I include some general comments or minor suggestions for changes to the basic reporting to improve readability and repeatability.

-1. BASIC REPORTING

Line 75-76: ‘Several studies…’ ends with no citations. It would be beneficial for the reader if some of these studies were referenced.
Citation of reviews: Preferably primary literature should be cited where a specific point of experimentation is being made, and therefore citation of review articles is not appropriate in all cases. For example: Lines 270 – 271 ‘our research…and that of others…’ cites two reviews that do not encompass research but summarise other findings. To both assist the reader in sourcing the appropriate research and to credit the prior researchers, the papers with the findings of importance should be cited. An alternative would be to state ‘reviewed in ….’, which could be used, for example, when introducing limitations in serological diagnostic development (lines 76 – 78).
Accession numbers (beginning line 105): It would be beneficial to note the database these accession numbers are referring to. I assume Genbank based on their structure, but adding ‘Genbank accession number’ at least once will improve clarity. In addition to this, the accession number for S. scabiei at line 105 seems out of place, as the authors go on to say they used 87 sequences, not just AB778896. Perhaps a supplementary table of all of the accession numbers used, and the host for the S. scabiei, would be helpful for reproducibility.
Supplementary Figure 1: Similar to the above query, it isn’t clear whether the haplotypes are produced by the authors or from another manuscript. My assumption is that the alignment of 87 sequences mentioned was reduced by the author to the 21 haplotypes displayed. If this is the case, I’d recommend reducing lines of the image further by combining sequence 1 (“ITS-2”), sequence 7 “Haplotype 6”, and sequence 23 “S. scabiei AB778896” into one ‘haplotype’ as they all appear to be the same sequence? And list the 87 accession numbers, along with host/country of origin, in another supplementary table.
Raw data: It is mentioned that sequencing was undertaken to confirm the amplification of the correct target (Line 126-127), and therefore to follow the data policy of PeerJ it would be useful to include this sequence data as a supplementary FASTA file or via submission to Genbank. Particularly as it is cited as evidence/validation that the LAMP has amplified the correct product.

2. EXPERIMENTAL DESIGN

• Original primary research within Scope of the journal.
• Research question well defined, relevant & meaningful. It is stated how the research fills an identified knowledge gap.
• Rigorous investigation performed to a high technical & ethical standard.
• Methods described with sufficient detail & information to replicate.
The work presented is original, providing the first available LAMP assay for an important pathogen and warrants publication. The assay is suitably validated using controls, confirmation that negatives were not due to inhibition and comparison to current standards (eg microscopy). I have only one suggestion I could consider a ‘major’ change and several minor suggestions to improve reproducibility and interpretability.
Major:
The final stage of the methodology (section 2.4) describes a rapid method of extraction and detection from skin scrapings, and compares the results to microscopy. However in the manuscript the authors note that microscopy is not as reliable as the PCR they had developed previously. Therefore reporting the results of the PCR assay of these samples would allow more robust comparison and a true measure of this rapid extraction technique. It is also not clear whether these samples come from individual animals and clarification is needed. Additionally as the microscopy is now mentioned as a core comparison tool, some additional details of methodology should be included so that the reader can more readily understand how samples are assessed.
Minor:
Lines 133 – 134 ‘Previously determined by PCR and microscopy’ requires a reference for the microscopy, and perhaps a ‘(below)’ for the PCR, as it isn’t clear to the reader at this point how this prior testing was undertaken.
Lines 137 – 139: It would benefit the reader, particularly those unfamiliar with Australian fauna (as the PeerJ reader may be), to have the scientific names of species listed, as well as full common names when first mentioned (eg. Domestic dog, bare-nosed wombat, etc).
Lines 105 – 106: For repeatability, briefly mention the algorithm used to generate the alignment (eg. Muscle, MAFFT, etc) as these can impact the final alignment.
Lines 105 – 108: Mention Figure S1 here for the reader.
Lines 127 – 129: A few more words of detail could be added to clarify that this process was used as the final means of running the assay on the test/clinical samples (rather than a basic heat block for the development). Also, broadly speaking (and perhaps based solely on personal experience) annealing in the molecular biology world is interpreted as primer annealing. Perhaps the sentence would be clearer if it was highlighted that this is a method of generating an annealing/melt curve based on the sequence of the amplified product.
Lines 139 – 140: It would be beneficial to at least mention the kit used for the extractions, as not all readers will have access to the cited manuscript.
Lines 145 – 146: The Victorian department was renamed The Department of Environment, Land, Water, and Planning in 2015 and may need to be updated in the manuscript accordingly. If not, the word ‘research’ should at least be removed from the title (that is, Department of Environment and Primary Industries)
Line 157: For clarity, please write out chemical name in full and include pH
Line 161: Change elute to aliquot (or another appropriate word), as elute implies there was a wash step in your extraction.
Supplementary table 2: For clarity, I recommend having the two koala samples side by side, rather than either side of the wombat/wallaby samples. Similarly, having all of the wombat samples together rather than separated by other species. If the order is based on another value that I’ve failed to identify please clarify.
Supplementary table 2: Perhaps rename the ‘374PCR’ column as cox1 PCR, as 374 is ambiguous when read in isolation.

3. VALIDITY OF THE FINDINGS

• Impact and novelty not assessed. Negative/inconclusive results accepted. Meaningful replication encouraged where rationale & benefit to literature is clearly stated.
• Data is robust, statistically sound, & controlled.
• Conclusions are well stated, linked to original research question & limited to supporting results.
• Speculation is welcome, but should be identified as such.

4. General comments

As no human samples were tested, I found the title slightly misleading. Whilst the research does focus on an assay for an ectoparasite of humans and animals, the title inherently implies (most likely unintentionally) that human samples were tested.
Introduction
Line 51. ‘Symptoms’ should be changed to ‘clinical signs of disease’ in this instance, as animals are being referred to. The term ‘symptoms’ should be reserved for subjective observations by a patient (e.g. ‘a sore throat’).
Line 58 – 60. Improve for clarity. In my mind if the organism is endemic in humans (rather than wildlife) then the spill over or spill back is occurring inversely to that described in the manuscript (that is, an organism endemic to humans would ‘spill over’ into an animal host).
Lines 76 – 78, Improve for clarity, ‘cross reactivity between mites’ may be better said along the lines of ‘cross reactivity between S. scabiei antigens and those from other mite species’, or something to that effect.
Line 98: Add ‘The’ to the start of the sentence ‘LAMP assay was evaluated…’ for improved readability
Throughout the manuscript plural acronyms (eg NAATs, ELISAs) are written with a possessive apostrophe (eg. NAAT’s, ELISA’s), which should be rectified.
Methods
Lines 109 and 115. Based on PeerJ author instructions, websites should be listed in the references section (as the authors have done for epitools.ausvet.com.au). Additionally, both the tools listed have citable publications which may alleviate the difficulty of citing websites in the final reference list (Johnson et al, 2008, NCBI BLAST: a better web interface, Nucleic Acids Research AND, Owczarzy et al, 2008, IDT SciTools: a suite for analysis and design of nucleic acid oligomers, Nucleic Acids Research)
References: Scientific names (eg. Chlamydia, Mycoplasma, etc) should be italicised throughout
Footnotes should be used for abbreviations in all tables, rather than using the title/legend section. In particular the use of NA is ambiguous and needs further clarification
Figure 1. ITS-2 gene should be noted in either the title or the legend for the figure to be ‘self contained’
Figure S2: The values of the serial dilutions should be adjusted for appropriate rounding. 2.8 should dilute to either 0.28 or 0.3, not 0.2. The same for 8.6 to 0.8. Perhaps a log adjusted value would be easier to read (eg -4.7 rather than 0.000002).
Table 1. The sensitivity/specificity confidence intervals should be written as percentages, not as proportions (that is 86-100, not 0.86-1.00).
Table S1. ‘Boop B’ should be Loop B
Table 2 and Table S2. Time values should either be separated by a colon ‘:’ (12:15) or written in minute/second format (that is 12’15”), rather than using decimal. Standard deviation should be ±, not just +.

Reviewer 1 ·

Basic reporting

-

Experimental design

-

Validity of the findings

-

Additional comments

The authors developed LAMP, as a part of their continue efforts in improving diagnosis of S. scabies infestation. The authors proposed that LAMP could serve an important role in a point-of-care or resource-limited setting. I have some comments, below:

> Line 83-85: please provide reference(s).

> Line 173: A total of 87 ITS2 sequences were aligned and shown to have 96.5 - 100% sequence identity. I wonder how much % identity we should expect to consider these sequences were from the organisms with the same species? Should that be 99-100%?

> Line 182, after incubation in a thermal block, the LAMP products were detectable, as visualised by the amplicons on the gel. I wonder what/how interpretation/determination of a LAMP result is relied on? Is it by looking directly at the reaction tube, observing an amplicon on a gel, or checking on a curve/peak (as shown in Fig 2)? A point of care test is expected an convenient way of obtaining and interpretting a result, and that should be described and evaluated in the study.

> The LAMP assay with a heat block (Line 180), a thermal cycler (Line 183), and a fluorometer (Line 186), I wonder which one showed the better detection performance? Comparing an effect of these incubation tools will be useful to readers who may have different devices available, or have to decide which device should be used.

> Figure S2, the marker is not clear, and not labeled.

> Line 194: The authors described the high congruence (95.3%) between LAMP and PCR, and showed that LAMP had higher detection sensitivity than conventional PCR. I feel it is bit confused about the Table 1, where there are the results of 64 clinical samples being compared. However, Line 136-138, the author mentioned that there were a total of 78 clinical samples, consisting of 40 positive samples (from 23 subjects) and 38 negative samples (from 24 subjects) (the authors should briefly described here how these samples were defined positive and negative, although a reference is referred to). Which set of samples (64 or 78 samples) was used to calculate the sensitivity and specificity? The detection sensitivity and specificity of the conventional PCR should be mentioned.

> Line 246-250, The higher sensitivity of LAMP compared to conventional PCR, how could the authors exclude a possibility of false positive result by LAMP? The authors should also simply perform, compare, and show an analytical sensitivity by using serial dilution samples to investigate which assay (LAMP vs PCR) could detect lower amount of a define DNA sample. In addition, how the authors manage to control an contamination issue of a high sensitivity assay like LAMP?

> The author emphasized an advantage of using LAMP over PCR by that LAMP could be used as a part of a point-of-care (POC) and resource-limited settings. However, it seemed that the results of LAMP were read on a gel, or read by a fluorometer, which may not be considered as a part of POC.

> Line 263-164, the possible targets for LAMP assay can be ITS-2, cox1, 12S rRNA, and 16s rRNA. I suggested that the authors incorporate the initial analysis to indicate that how and why ITS-2 was selected as the target for LAMP, i.e., how much sequence variation, how much %GC, etc.

> Not only detection efficiency, some other factors are important in determining which assay should be selected for use are cost, time, and personal skill. Please describe these factors, for LAMP in comparison with other assays (PCR, microscopy).

Reviewer 2 ·

Basic reporting

a. The introduction provides context, but is lacking information on other assays that are present. It also does not describe limitations to the assay, and why those limitations are tolerable in the context for this assay development
b. Would like to see Figure S2 and Table S1 included in the main body of the text as opposed to supplementary files

Experimental design

a. The scope of the research and trials appear to be relevant for the journal.
b. More information about the materials and methods should be described, as there are some significant portions that are needed to describe the methods. Please see the General Comments below for a line-by-line description.
c. The objectives and hypotheses for the study were not clearly defined in the introduction. Would like to see them explicitly stated to allow for better flow of research description.
d. There were really very few samples actually tested in the survey of PCR to LAMP samples. It would be nice to see more samples assessed, but there are of course limitations to the number of samples that one can obtain.

Validity of the findings

a. Certainly a good body of work that could be a useful diagnostic tool, especially in lower-income countries.
b. It would be nice to see more discussion on the limitations and benefits of this amplification method over others. There are many new developments that would improve the assay developed here, and there are certainly pitfalls to a LAMP assay that should be discussed.

Additional comments

a. The abstract could be abbreviated to be more concise and have no separated sections, which would make it easier to read.
b. Line 85: You should discuss other diagnostic tools. For example, Biomeme has an iPhone qPCR device and Agdia has a portable RP device. It would be nice to see a comparison of methods for this system.
c. Lines 86-94: There’s several limitations to LAMP as opposed to other methods, especially real-time methods. Turbidity (or lack-thereof) can be misleading in results, whereas a PCR product is pretty distinct. Comment on some of the limitations, and where your methodology accounts for those limitations.
d. Line 94: It would also be nice to see any comparisons to other LAMP/Nucleic Acid Assays developed for micro-scale arthropods. Are there any limitations to working with those systems (ie PCR/LAMP inhibitors) that may cause issues?
e. Line 95-101: This seems more appropriate for the abstract. You should list your hypotheses/objectives for the study here as opposed to summarize the work as you’ve already done in the abstract.
f. Line 104: How many copies of the target region are found in one mite? The LAMP assay can amplify as few as 6, but can the assay detect less than one mite? (ie, should there be broken up mite from the sampling process, could the assay still detect it?)
g. Line 119: LF and LB are described here but not in the primer design above. Are these loop primers? This should be described in the primer design as well (lines 104-116).
i. Line 124-125: This should be included in the primer design section.
h. Line 131: Was sensitivity not assessed during LAMP assay development?
i. It would probably make more sense to swap the Clinical samples discussion and the DNA extraction methodology since the sample information describes that DNA extraction was conducted.
j. Line 153: I’m confused as to what you’re testing against for inhibition. Within the skin samples? Likely inhibitors would also be present in the mite only DNA samples as well. Would it not have been better to create a plasmid or an internal control to look at inhibition?
k. Line 155: Again, I think the DNA extraction should be described before clinic samples. Even if it is described in another paper, a quick overview of the process would be nice in the assay development.
l. Line 159: Lyse’n’LAMP was not specified in the previous assay development description. Should that not also be mentioned there? Also, why did you switch master mix here? Were there issues with the regular Optigene master mix?
m. Line 161: “aliquot” instead of “elute”?
n. Line 164: Please go into more detail as to how this is calculated and why this was chosen as opposed to ROC statistics and/or chi-square/fisher’s exact that are typically used to compare assays. This is not a common statistic, so it should be given more detail.
o. Line 172: Did you extract DNA from these sources and run the LAMP reaction with those extracts as well? In silico is fine, but an actual assessment with primers under reaction conditions may have differential binding than what is predicted, especially if the reaction temperature is not optimal for the primer sets used.
i. Found this below (189). Please specify the species assessed in vitro and number of replications. This should be described in the materials and methods as well.
p. Line 180: Was this detectable by precipitate or using the assay parameters intended for the POC?
q. Line 184: The sensitivity of the assay wasn’t well described in the materials and methods. I would explicitly describe conditions for this assessment and how many reactions were run with the dilution series.
r. Line 206: Inhibition of the assay may not be necessarily no-amplification with a LAMP assay given the high affinity for the target region to amplify with the Bst polymerase. It could just delay the amplification significantly to look like non-specific binding. That would be more evident a real-time reaction. There could still be inhibition and amplification taking place. I would rephrase that the potential inhibitors in the system did not prevent amplification from occurring.
s. Line 214: Please describe this in better detail in the materials and methods
t. Line 230: qPCR methods and RP methods are capable of this as well, why is this better than those methods?
u. Line 230: Could you describe the potential minimal amount of DNA present in a typical skin scraping? It may help to describe the need for a lower limit. For example, if there should be less than a mite in a given sample due to poor collection or low colonization of a lesion, would the assay be able to detect ½ a mite or less worth of DNA?
v. Line 245-250: LAMP can amplify as few as 6 copies of a target region in a given reaction whereas PCR requires more DNA to amplify, so your increased sensitivity is not uncommon and a large appeal for the assay in low-DNA assessments.
w. Line 253: If you want to discuss the DNA extraction in the discussion, it really should be described in the materials and methods. It seems that it would be part of the novelty of the assay and largely part of the whole process for a limited-technology facility.
x. There should be discussion about Line 211 findings. The high variability in the real-time amplification of LAMP products: could this be due to inhibitors present from the crude DNA extraction process? Variability of LAMP product amplification should be discussed since this is not the only occurrence of it in the literature.
y. Line 277-278: The cost of the equipment is a rapidly changing environment, especially with those using Genie equipment. It’s almost as expensive as some real-time PCR machines. It may be worth elaborating on other diagnostic methods available and why the LAMP assay may be better to pursue in this setting. An actual breakdown of costs does not necessarily show that LAMP is the most inexpensive tool unless crude tools (heat block with visual detection of precipitate) are used.

---

## Round 0.2 · Minor Revisions

Please, include some information regarding validity of the findings suggested by reviewer 2.

Reviewer 1 ·

Basic reporting

-

Experimental design

-

Validity of the findings

-

Additional comments

The authors have addressed all of my concerns.

Reviewer 2 ·

Basic reporting

Article follows basic reporting required by journal.

Experimental design

Research questions defined and define research gap. Reviews about experimental design were addressed.

Validity of the findings

Testing was very limited, and few samples were assessed for sensitivity and specificity of assay. The authors describe the limitation of this in the discussion. There is also no sequencing of LAMP products to confirm non-specific binding or primer dimerization. If this work was done, a statement of that work should be included.

---

## Round 0.3 · accepted · Accept

Congratulations for this interesting manuscript. This diagnostic method will be very useful for detecting S scabei in animals; in most domestic and wild animals it is considered as a relevant and common disease with potential capacity to infect humans.

#